# Accuracy of a Prehospital Triage Protocol in Predicting In-Hospital Mortality and Severe Trauma Cases among Older Adults

**DOI:** 10.3390/ijerph20031975

**Published:** 2023-01-20

**Authors:** Axel Benhamed, Marcel Emond, Eric Mercier, Matthieu Heidet, Tobias Gauss, Pierre Saint-Supery, Krishan Yadav, Jean-Stéphane David, Clement Claustre, Karim Tazarourte

**Affiliations:** 1Service SAMU-Urgences, Centre Hospitalier Universitaire Édouard Herriot, Hospices Civils de Lyon, 69123 Lyon, France; 2Centre de Recherche, CHU de Québec-Université Laval, Québec, QC G1J 1Z4, Canada; 3SAMU 94, Hôpitaux Universitaires Henri Mondor, Assistance Publique-Hôpitaux de Paris (AP-HP), 75610 Paris, France; 4Anaesthesia Critical Care, Grenoble Alpes University Hospital, 38700 Grenoble, France; 5Department of Emergency Medicine, University of Ottawa, Ottawa, ON K1N 6N5, Canada; 6Clinical Epidemiology Program, Ottawa Hospital Research Institute, Ottawa, ON K1Y 4E9, Canada; 7Service d’Anesthésie-Réanimation, Centre Hospitalier Universitaire Lyon Sud, Hospices Civils de Lyon, 69310 Pierre-Bénite, France; 8Research on Healthcare Performance (RESHAPE), INSERM U1290, Université Claude Bernard Lyon 1, 69100 Lyon, France; 9RESUVal Trauma Network, Centre Hospitalier Lucien Hussel, 38200 Vienne, France

**Keywords:** trauma, emergency medical services, triage, older adults

## Abstract

*Background*: Prehospital trauma triage tools are not tailored to identify severely injured older adults. Our trauma triage protocol based on a three-tier trauma severity grading system (A, B, and C) has never been studied in this population. The objective was to assess its accuracy in predicting in-hospital mortality among older adults (≥65 years) and to compare it to younger patients. *Methods*: A retrospective multicenter cohort study, from 2011 to 2021. Consecutive adult trauma patients managed by a mobile medical team were prospectively graded A, B, or C according to the initial seriousness of their injuries. Accuracy was evaluated using sensitivity, specificity, positive and negative predictive values, and positive and negative likelihood ratios. *Results*: 8888 patients were included (14.1% were ≥65 years). Overall, 10.1% were labeled Grade A (15.2% vs. 9.3% among older and younger adults, respectively), 21.9% Grade B (27.9% vs. 20.9%), and 68.0% Grade C (56.9% vs. 69.8%). In-hospital mortality was 7.1% and was significantly higher among older adults regardless of severity grade. Grade A showed lower sensitivity (50.5 (43.7; 57.2) vs. 74.6 (69.8; 79.1), *p* < 0.0001) for predicting mortality among older adults compared to their younger counterparts. Similarly, Grade B was associated with lower sensitivity (89.5 (84.7; 93.3) vs. 97.2 (94.8; 98.60), *p* = 0.0003) and specificity (69.4 (66.3; 72.4) vs. 74.6 (73.6; 75.7], *p* = 0.001) among older adults. *Conclusions*: Our prehospital trauma triage protocol offers high sensitivity for predicting in-hospital mortality including older adults.

## 1. Introduction

Older adults represent an expanding segment of the trauma population within developing countries’ healthcare systems; their proportion has increased from 18% in 2005 to 30% in 2015 in the US [1], with a similar trend in Europe [2,3]. Despite this trend, advanced age has been consistently associated with higher proportions of undertriage in the prehospital setting [4,5,6,7]. Although undertriage rates vary across studies, it can reach 50% [8] and patients older than 75 with an injury severity score (ISS) of more than 15 have been found to be 50% less likely to be triaged to a major trauma center compared to their younger counterparts [9]. The American College of Surgeons Committee on Trauma (ACSCOT) has emphasized a need for senior-tailored care [10], but much of that focus has concerned their care once they reach the hospital. Indeed, the identification of older adults with severe trauma as well as the setup of direct access to specialized trauma centers remain suboptimal [11,12,13]. However, the importance of prehospital triage in trauma care cannot be overstated. An accurate prehospital protocol is the cornerstone of a mature trauma system and transport to the most suitable center with a high level of trauma care designation is often regarded as the best option since such centers are associated with a lower risk of death and reduced morbidity [14,15,16]. Undertriage may be partially attributable to the inaccurate recognition of seriously traumatized older adults. Indeed, two recent systematic reviews suggested that current prehospital trauma triage tools may be at fault, as they do not accurately identify older patients with serious injuries, and undertriage is still an ongoing issue [13,17]. This is a critical concern because admission of severely injured older adults to healthcare facilities with higher levels of trauma care designation is associated with gains in the probability of survival [18]. Therefore, the crucial need to focus future research on the development of specific silver triage criteria has been recently highlighted [19,20,21].

Within the French emergency medical services (EMS) system, prehospital triage is performed by physicians, and in some regions, the triage protocol is based on a three-tier severity grading system that integrates clinical evaluation, response to prehospital resuscitation, and trauma circumstances as a surrogate for trauma severity [22]. A recent meta-analysis found that this protocol had high accuracy in predicting an ISS of more than 15, mortality within 30 days, or admission to intensive care unit (ICU) [23]. However, the accuracy of this triage protocol in specifically identifying severely injured older adults has never been evaluated. 

Hence, we aimed to evaluate the accuracy of this triage protocol in predicting mortality among older adults (≥65 years) and compare it to younger patients. The secondary objective was to evaluate its accuracy in predicting severe trauma cases.

## 2. Materials and Methods

### 2.1. Study Design and Setting

This retrospective multicenter cohort study used data from the *Réseau des Urgences de la Vallée du Rhône* (RESUVal) Trauma Registry. The registry covers an area of approximately 3 million inhabitants and includes 3 level-I (one pediatric trauma), 1 level-II, and 5 level-III trauma centers. RESUVal prospectively collects pre- and in-hospital information (trauma room, emergency department (ED), and intensive care unit (ICU) if applicable) on all consecutive trauma patients managed by a mobile medical team (MMT).

The French prehospital EMS has been described previously [7,24,25,26]. Briefly, it is a 24 h physician-led system, and out-of-hospital suspected severe life-threatening trauma situations are managed by the *Service d’Aide Médicale d’Urgence* (SAMU). Country-wide 24/7 access to the SAMU is provided by a single national telephone number (15) where a dispatching physician assesses the situation and can activate either a helicopter or ground MMT in suspected critical cases. The MMT is composed of a physician, a nurse, an ambulance driver, and a medical resident (in academic centers). Patient orientation depends on a regional triage protocol (Appendix A) based on a 3-tier grading system. It evaluates the seriousness of patient injuries at presentation on scene and the response to treatment during prehospital resuscitation, and integrates physiological information, trauma circumstances, and comorbidities. A patient is prospectively labeled by the on-scene physician as a Grade A (unstable despite resuscitation), B (stabilized after prehospital resuscitation), or C (stable with high-kinetic circumstances or specific medical conditions) major trauma (Figure 1).

### 2.2. Study Population

All consecutive trauma patients aged 18 years or older managed by a prehospital MMT from January 2011 to December 2021 were considered for analyses. Those who died in the prehospital setting (either at the scene or during transportation) or with missing data pertaining to age, trauma grade, and/or ISS were excluded. Older adults were defined as those aged ≥65 years. Analyses were also performed within three subgroups among older adults (65–74, 75–84, and ≥85 years).

### 2.3. Data Collection

Prehospital, ED, and ICU data are collected by the physician in charge of the patient, while research technicians provide continuous monitoring of the completeness and correctness of the registry. They also collected patient outcomes at hospital discharge. Data management and analyses are performed by a data-manager-statistician (CC), from RESUVal. MMT physicians are asked to fill out a standardized case report form for any trauma patient with at least one Vittel criterion corresponding to effective or suspected seriously injured patients [27], and a research assistant reviews patients’ medical records in case of an incomplete case report form.

### 2.4. Study Data

The following prehospital variables were prospectively recorded: sociodemographic and trauma characteristics, first physiological parameters measured by the MMT (including point-of-care (POC) capillary hemoglobin concentration), and data pertaining to prehospital MMT advanced life support (ALS) procedures. The Abbreviated Injury Scale (AIS, based on the 1998 version) and ISS were calculated after anatomical and physiological assessments were completed.

### 2.5. Outcome Measures

The main outcome was the accuracy of the triage tool in predicting in-hospital mortality. Accuracy metrics were the sensitivity (Se), specificity (Sp), negative (NPV) and positive predictive value (PPV), and negative (NLR) and positive (PLR) likelihood ratio. The secondary outcome tested was the accuracy in predicting severe trauma cases defined as those with an ISS >15.

### 2.6. Statistics

Categorical data were described by frequencies and proportions (%), and continuous data were described by medians and interquartile ranges (IQR), first and third quartiles]. Univariable statistical comparisons of continuous data were performed using the nonparametric Wilcoxon rank sum test for two-group comparisons and the Kruskal–Wallis rank sum test for three-group comparisons. The Pearson chi-squared test was used for categorical data. For each outcome, Se, Sp, NPV, PPV, NLR, and PLR were computed using the epiR library [28].

Considering that all patients were graded A, B, or C, we did not test grade C accuracy as it would have led to 100% sensitivity. Consequently, Grade A and Grade B were considered thresholds for the different outcomes. Because the triage protocol is expected to predict severe trauma cases to avoid undertriage, we focused on presenting Se and NPV. The comparisons of accuracy metrics between groups were realized using the Pearson chi-squared test. Missing data were not imputed. We conventionally accepted a p value less than 0.05 as statistically significant. Statistical analyses were performed by a statistician (CC) using R^®^ software (version 4.1.2).

### 2.7. Ethics Approval and Consent to Participate

All patients received written information on their information being used for research and could oppose the use of their data. Given that the study was retrospective in nature and that patient information was anonymized before the analysis, the need for an ethics committee as well as written consent was waived according to French law. The study received approval from the national data protection agency (Commission Nationale de l’Informatique et des Libertés, CNIL; DE-2012-059) and the Advisory Committee on the Treatment of Research Information (*Comité consultatif sur le traitement de l’information en matière de recherche*, CCTIRS). All methods were performed in accordance with the Declaration of Helsinki.

## 3. Results

### 3.1. Patient Characteristics

During the study period, a total of 8888 patients were included (Appendix A), of whom 1250 (14.1%) were aged ≥65 years. Older adults were less frequently male than younger adults (63.6% vs. 77.4%, *p* < 0.0001), and the main trauma mechanism was a road traffic accident in both age groups (49.2% and 64.0%, *p* < 0.0001). At MMT arrival in the prehospital setting, the proportion of patients with hypoxia (SpO2 < 95%; 29.3% vs. 14.4%), anemia (hemoglobin < 9 g/dL; 4.3% vs. 2.0%), or decreased level of consciousness (GCS ≤ 8; 19.0% vs. 11.9%) was higher among older adults (*p* < 0.0001). Appendix A reports patient characteristics in the subgroups of patients aged 65–74, 75–84, and ≥85 years.

A total of 10.1% of patients were labeled Grade A (15.2% vs. 9.3% among older and younger adults, respectively), 21.9% were Grade B (27.9% vs. 20.9%), and 68.0% were Grade C (56.9% vs. 69.8%). Overall, most patients were admitted to a level-I trauma center. The median (IQR) ISS was higher among older adults (17 (9–25) vs. 11 (5–22), *p* < 0.0001, Table 1).

### 3.2. Accuracy of the Triage Protocol in Predicting In-Hospital Mortality

A total of 7.1% (*n* = 571) of patients died (19.8% vs. 5.0% among older and younger adults, respectively, *p* < 0.0001). Mortality was significantly higher among older adults at all severity grades (Grade A: 65.3% vs. 40.6%, Grade B: 28.7% vs. 5.7%, and Grade C: 3.6% vs. 0.2, *p* < 0.0001, Table 2).

Grade A sensitivity (%) (CI95%) in predicting mortality was significantly lower in the group of older adults (50.5 (43.7; 57.2) vs. 74.6 (69.8; 79.1), *p* < 0.0001). Similarly, Grade A NPV (%) (CI95%) was lower among older adults (88.4 (86.2; 90.4) vs. 98.6 (98.3; 98.9), *p* < 0.0001). Grade B was associated with higher sensitivity than Grade A but was still lower among older adults (89.5 (84.7; 93.3) vs. 97.2 (94.8; 98.6), *p* = 0.0003). Similarly, NPV was lower in older adults labeled grade B (96.4 (94.7; 97.7) vs. 99.8 (99.6; 99.9), *p* < 0.0001, Table 3).

Grade A and B accuracy among the group of older adults (65–74, 75–84, and ≥85 years) are presented in Appendix A.

### 3.3. Accuracy of the Triage Protocol in Predicting Patients with an ISS >15

Overall, 42.6% (*n* = 3785) of patients had an ISS >15 (57.0% vs. 40.2% among older and younger adults, respectively, *p* < 0.0001). The proportion of patients with an ISS >15 was not significantly different between Grade A older and younger adults (87.9% vs. 89.8%, *p* = 0.27). Conversely, it was higher among older adults in the two other severity groups (Grade B: 74.8% vs. 67.6%, *p* = 0.01 and Grade C: 39.9% vs. 25.5%, *p* < 0.0001, Table 2).

Grade A sensitivity in predicting patients with an ISS >15 was not significantly different between the two age groups (23.5 (20.4; 26.7) vs. 20.7 (19.3; 22.2), *p* = 0.1161), but its NPV was significantly lower among older adults (48.6 (45.5; 51.6) vs. 64.8 (63.7; 66.0), *p* < 0.0001). Grade B showed higher sensitivity than Grade A, and it was significantly higher among older adults (60.1 (56.4; 63.7) vs. 55.8 (54.0; 57.5), *p* = 0.0393). Conversely, NPV associated with Grade B was significantly lower among older adults (60.1 (56.4;63.7) vs. 74.5 (73.3; 75.7), *p* < 0.0001; Table 3). Grade A and B accuracy among the group of older adults (65–74, 75–84, and ≥85 years) are presented in Additional file 4.

## 4. Discussion 

The A/B/C prehospital triage tool showed high sensitivity in predicting in-hospital mortality across all age groups but performed less well for older adults. A nonnegligible proportion of Grade C patients, those with an initial reassuring clinical exam but high-kinetic trauma circumstances or specific medical conditions, were found to be severe trauma cases, but very few of them died.

Given that identification of trauma severity at the scene directly influences patient triage and orientation, which may impact trauma-related mortality, having efficient triage protocols remains a cornerstone of any mature trauma system. This is particularly true for older adults. Early identification of major injuries in this population has been defined as the first decision-making step in managing older patients with injuries [20]. The triage protocol used herein demonstrated great accuracy in identifying the most severe cases, those who died during their in-hospital stay, regardless of age. Nevertheless, it was less efficient in identifying patients with an ISS >15. In another similar French prehospital trauma setting, using the same triage protocol, Bouzat et al. uniformly found a total of 89% of patients with an ISS >15 among Grade A (vs. 89.4% herein), 64% (vs. 68.9%) among Grade B, and 31% (vs. 27.2%) among Grade C patients [22]. The study neither provided diagnostic performance data, nor performed age stratification. In Canada, Coulombe et al. specifically assessed the accuracy of a five-step paramedic-led prehospital trauma triage protocol to identify the need for advanced trauma care in the older adults. They concluded that there was insufficient sensitivity (65.8% (95% CI: 48.7–80.4%)) when combining the five steps for patients aged ≥65 years, and this sensitivity was even lower among the oldest adults [29]. In the Netherlands, Voskens et al. assessed the performance of the national triage protocol, regardless of the actual destination facility, and reported an undertriage of 63.8% 95% CI (59.2–68.1%) [30]. More generally, it has been reported that almost all protocols have a low sensitivity, thereby failing to identify severely injured patients [31]. Nevertheless, sensitivity and specificity are incidence-dependent; therefore, the comparison of metric characteristics between studies should be interpreted with caution. Interestingly, Nishijima et al. assessed the performance of the US field triage guidelines for specifically identifying traumatic intracranial hemorrhage among older adults. They reported a very low sensitivity of 19.8% 95% CI (5.5–51.2%) [32]. These findings suggest that prehospital EMS should develop and use age-specific trauma triage criteria that should focus on the premorbid status of the individual as suggested before [21].

Despite the good performance of the grading system, the present study raises a serious point of concern. Among the group of adults graded C, mortality was less than 1%. However, a nonnegligible proportion of those patients who presented with an ISS >15 (more than one in four cases) and/or required advanced trauma interventions; these proportions were even higher in the group of older adults. Therefore, these findings invite in-hospital clinicians of level-III trauma centers to remain vigilant for Grade C patients. Indeed, based on our triage protocol, those potentially severely injured patients may be assigned to level-III healthcare facilities although fewer trauma care resources are available. Avenues to improve the diagnostic performance of Grade C and better identify and triage severely injured patients could be to consider removing high energy transfer criteria from the list. Accordingly, Cassignol et al. reported that no criterion related to kinetic elements was significantly correlated with an ISS >15, mortality within 30 days, or admission to the ICU [33]. Another possibility to improve the on-field identification of severe trauma cases among Grade C patients would be to add easy-to-use and quickly measurable clinical variables such as the shock index, which may help to improve the identification of severe cases. As such, substituting the criterion of SBP <90 mm Hg in the American National Trauma Triage Protocol with a shock index >1 resulted in a considerable reduction in undertriage (−5.9%) without a substantial rise in overtriage (+1.3%) [34]. Technologies such as capillary hemoglobin concentration measurement at the scene to screen for occult anemia are also an easy-to-use tool for physicians as well as paramedics in their daily routine and could be of help in discriminating severe cases. Some authors showed that hemoglobin variation (measured with a POC device) was able to predict significant hemorrhage [35]. Elsewhere, it has been suggested that integrating prehospital serum lactate measurement may improve the prediction of in-hospital mortality, emergent surgery, and multiple organ dysfunction syndrome [36,37]. More specifically, St John et al. showed that prehospital lactate was predictive of the need for resuscitative care among normotensive trauma patients [38], which could be of noteworthy help within the group of patients meeting only the high-kinetic trauma circumstances red flag. Although the current data showed feasibility, further research is still needed to establish whether lactate might truly provide meaningful guidance during prehospital triage [39]. Another insufficiently explored track would be to evaluate the accuracy of physician judgment regarding trauma severity when facing a hemodynamically stable patient presenting with only high-kinetic trauma circumstances. Indeed, although it has been shown elsewhere that the judgment of nonphysician EMS providers was useful in identifying high-risk patients missed by other criteria [40], the accuracy of physician judgment has never been investigated for this older adult trauma population.

Among the specific group of older adults, the accuracy of field triage protocols could be improved by integrating geriatric-specific criteria. For instance, adding preinjury comorbidities and geriatric-specific physiological measures has shown high sensitivity improvement at the cost of a decrement in specificity in identifying severe trauma (ISS ≥16 or need for major nonorthopedic surgery) [41]. 

A major strength of this study is that it was conducted in a wide area, including urban as well as rural zones, and patients were included over a large period, reducing selection bias, and assuring a high level of generalizability to comparable settings. Another strength is that the trauma registry captures most severe trauma cases given that very few patients are directly admitted to an ED by relatives or bystanders. However, the study suffers from some limitations. First, due to the observational retrospective nature of the study, potential confounding factors may have impacted comparisons between the two age groups. For instance, data concerning the decision to withhold and terminate resuscitation were not collected, which could have influenced the results related to grade performance, especially those regarding in-hospital mortality. Second, we do not have all data on the appropriateness of severity grade category assignment (e.g., some clinical findings pertaining to Grade B category or details on trauma mechanism pertaining to Grade C category were not collected in the registry). Third, it must also be acknowledged that the limited sample size of the very old patient groups can affect the interpretation of our results. Fourth, given that this study was conducted in a physician-staffed prehospital EMS, this limits the opportunity to compare the findings to other prehospital EMS, which are paramedic-led in most EMS [42].

Future research is warranted to identify the underlying reasons for the disparity in accuracy performance between the two age groups and to adjust the grading system criteria to population-specific characteristics of older adults to optimize the identification of high-risk patients. Another avenue for future studies based on large cohorts would be to integrate machine learning to enhance triage accuracy. Some authors have already developed an artificial-intelligence-based method for predicting various outcomes including shock, major surgery, and early massive transfusion in patients with truncal gunshot wounds [43]. Such a strategy may greatly help clinicians in facilitating patient risk assessment and triaging and optimize resource use. Elsewhere, machine-learning-based models predicted the need for urgent neurosurgery [44] or trauma mortality [45]. The potential clinical benefits of such models as decision-making and triage tools deserve further assessment in the prehospital environment. In addition, these models were built a posteriori. Therefore, computerization of prehospital EMS would be required to integrate such AI-based tools into daily healthcare professionals’ routines.

## 5. Conclusions

This prehospital triage protocol offers high sensitivity in predicting in-hospital mortality including in older adults, which is of great interest for prehospital and in-hospital clinicians, and for emergency service organization. Nevertheless, it needs to be refined to better identify severe trauma cases in the prehospital setting among patients presenting with reassuring initial clinical examinations and high-kinetic trauma circumstances or specific medical conditions.

## Figures and Tables

**Figure 1 ijerph-20-01975-f001:**
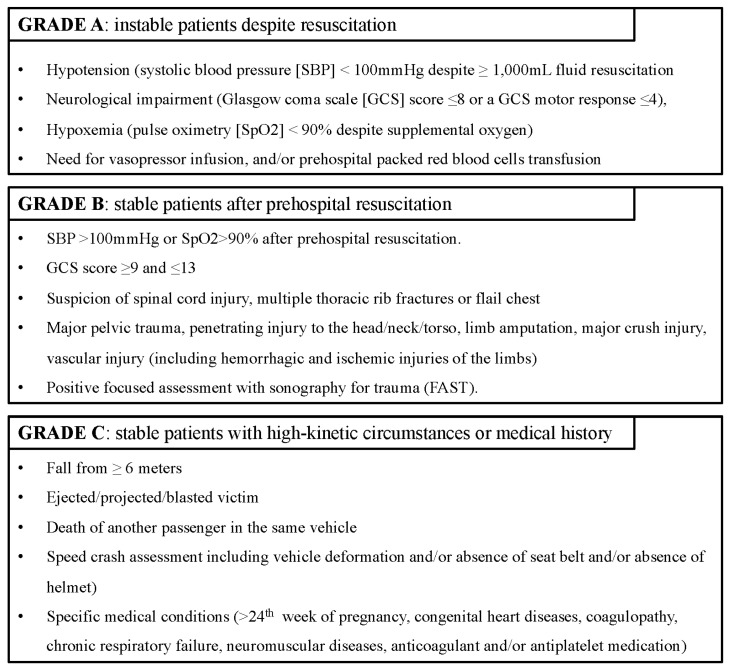
Major trauma grading system.

**Table 1 ijerph-20-01975-t001:** Patient characteristics.

	All Patients *n* = 8888	18–64 Years *n* = 7638	≥65 Years *n* = 1250	*p* Value
Age	38 (26–55)	34 (24–48)	74 (69–82)	<0.0001
Sex, male, *n* = 8857	6681 (75.4)	5890 (77.4)	791 (63.6)	<0.0001
Type of trauma, *n* = 8797				0.0227
Blunt	7959 (90.5)	6823 (90.2)	1136 (92.3)	
Penetrating	838 (9.5)	743 (9.8)	95 (7.7)	
Mechanism, *n* = 8807				
Road collision	5452 (61.9)	4844 (64.0)	608 (49.2)	<0.0001
Fall	1930 (21.9)	1486 (19.6)	444 (36.0)	<0.0001
Gunshot, stab	680 (7.7)	609 (8.0)	71 (5.8)	0.004
Other	745 (8.5)	633 (8.4)	112 (9.1)	0.44
Delays				
On-scene time, *n* = 6891	33 (25–45)	33 (24–45)	35 (25–46)	0.0245
Total prehospital time ^a^, *n* = 5668	80 (60–105)	79 (60–104)	85 (65–111)	0.0001
Prehospital medical evaluation				
Systolic blood pressure, *n* = 8140	129 (113–143)	127 (112–140)	140 (120–163)	<0.0001
Heart rate, *n* = 8190	88 (75–101)	89 (76–103)	82 (70–96)	<0.0001
SpO2 <95%, *n* = 7427	1219 (16.4)	920 (14.4)	299 (29.3)	<0.0001
Hemoglobin <9 g/dL, *n* = 7378	172 (2.3)	129 (2.0)	43 (4.3)	<0.0001
Shock index >1, *n* = 7903	764 (9.7)	691 (10.2)	73 (6.6)	0.0003
GCS score ≤8, *n* = 8155	1051 (12.9)	834 (11.9)	217 (19.0)	<0.0001
Prehospital ALS procedures				
Mechanical ventilation, *n* = 7899	1738 (22.0)	1431 (21.0)	307 (28.0)	<0.0001
Thoracostomy, *n* = 7841	90 (1.2)	81 (1.2)	9 (0.8)	0.3628
Packed red blood cell transfusion, *n* = 8022	128 (1.6)	107 (1.6)	21 (1.9)	0.4826
Fluid resuscitation, *n* = 8066	6366 (78.9)	5500 (79.2)	866 (77.4)	0.1883
Vasopressor infusion, *n* = 8030	707 (8.8)	567 (8.2)	140 (12.6)	<0.0001
Tranexamic acid, *n* = 6882	3569 (51.9)	3048 (51.5)	521 (53.8)	0.1993
Pelvic binder, *n* = 7843	1279 (16.3)	1154 (17.1)	125 (11.5)	<0.0001
FAST, *n* = 7770	2626 (33.8)	2331 (34.8)	295 (27.5)	<0.0001
Anticoagulant and/or antiplatelet medication	636 (7.7)	199 (2.8)	437 (37.1)	<0.0001
Severity grade				<0.0001
A	898 (10.1)	708 (9.3)	190 (15.2)	
B	1944 (21.9)	1595 (20.9)	349 (27.9)	
C	6046 (68.0)	5335 (69.8)	711 (56.9)	
Trauma center admission level				0.3113
I	7529 (84.7)	6483 (84.9)	1046 (83.7)	
II	756 (8.5)	633 (8.3)	123 (9.8)	
III	602 (6.8)	521 (6.8)	81 (6.5)	
Compliance to the field triage protocol ^b^	8823 (99.3)	7587 (99.3)	1236 (98.9)	0.1186
ISS	13 (5–24)	11 (5–22)	17 (9–25)	<0.0001
In-hospital trauma procedures ^c^				
Packed red blood cell transfusion	897 (10.1)	704 (9.2)	193 (15.4)	<0.0001
Vasopressor infusion	1187 (13.4)	943 (12.4)	244 (19.5)	<0.0001
Surgery, *n* = 8836	2587 (29.3)	2322 (30.6)	265 (21.3)	<0.0001
Angioembolization, *n* = 8834	270 (3.1)	220 (2.9)	50 (4.0)	0.0396
Chest tube drainage	495 (5.6)	435 (5.7)	60 (4.8)	0.2252
ICU admission, *n* = 8711	4008 (46.0)	3248 (43.4)	760 (62.0)	<0.0001

Data are presented as median (IQR) or n (%); proportions (%) were calculated among those with data. ^a^ From call to hospital arrival, expressed in minutes. ^b^ Defined as the proportion of patients admitted to a trauma center whose level of care is in accordance with the triage protocol. ^c^ Within the first 24 h. ALS: Advanced life support; FAST: Focused assessment with sonography for trauma; GCS: Glasgow coma scale; SpO2: Peripheral oxygen saturation measured by pulse oximetry.

**Table 2 ijerph-20-01975-t002:** Outcomes.

	All Patients *n* = 8888	18–64 Years *n* = 7638	≥65 Years *n* = 1250
Severe trauma as per the composite definition	4294 (48.3)	3477 (45.5)	817 (65.4)
Grade A	888/898 (98.9)	701/708 (99.0)	187/190 (98.4)
Grade B	1739/1944 (89.5)	1407/1595 (88.2)	332/349 (95.1)
Grade C	3293/6046 (54.5)	2839/5335 (53.2)	454/711 (63.9)
ISS >15	3785 (42.6)	3073 (40.2)	712 (57.0)
Grade A	803/898 (89.4)	636/708 (89.8)	167/190 (87.9)
Grade B	1339/1944 (68.9)	1078/1595 (67.6)	261/349 (74.8)
Grade C	1643/6046 (27.2)	1359/5335 (25.5)	284/711 (39.9)
In-hospital urgent and specialized trauma care ^a^	5554 (62.5)	4639 (60.7)	915 (73.2)
Grade A	877/908 (96.6)	694/708 (98.0)	183/200 (96.3)
Grade B	1687/1944 (86.8)	1363/1595 (85.5)	324/349 (92.8)
Grade C	2990/6046 (49.5)	2582/5335 (48.4)	408/711 (57.4)
In-hospital mortality	571 (7.1)	351 (5.0)	220 (19.8)
Grade A	373/816 (45.7)	262/646 (40.6)	111/170 (65.3)
Grade B	165/1673 (9.9)	79/1373 (5.7)	86/300 (28.7)
Grade C	33/5593 (0.6)	10/4950 (0.2)	23/643 (3.6)

Data are presented as n (%); proportions (%) were calculated among those with data. ISS: Injury severity score. ^a^ Includes the need for ≥1 of the following urgent and specialized in-hospital trauma care: surgery and/or angioembolization within 24 h after admission, chest tube drainage, endotracheal intubation with mechanical ventilation, vasopressor infusion, packed red blood cell transfusion, and/or ICU admission.

**Table 3 ijerph-20-01975-t003:** Performance of the grading system to predict in-hospital mortality.

		In-Hospital Mortality		
		18–64 Years	≥65 Years	*p* Value
Grade A	Se	74.6 (69.8; 79.1)	50.5 (43.7; 57.2)	<0.0001
	Sp	94.2 (93.6; 94.7)	93.4 (91.6; 94.9)	0.3776
	PPV	40.6 (36.7; 44.5)	65.3 (57.6; 72.4)	<0.0001
	NPV	98.6 (98.3; 98.9)	88.4 (86.2; 90.4)	<0.0001
	PLR	12.9 (11.5; 14.4)	7.6 (5.8; 10.1)	0.00007
	NLR	0.27 (0.22; 0.32)	0.53 (0.46; 0.61)	<0.0001
Grade B	Se	97.2 (94.8; 98.6)	89.5 (84.7; 93.3)	0.0003
	Sp	74.6 (73.6; 75.7)	69.4 (66.3; 72.4)	0.001
	PPV	16.9 (15.3; 18.6)	41.9 (37.4; 46.5)	<0.0001
	NPV	99.8 (99.6; 99.9)	96.4 (94.7; 97.7)	<0.0001
	PLR	3.8 (3.7; 4.0)	2.9 (2.6; 3.3)	<0.0001
	NLR	0.04 (0.02; 0.07)	0.15 (0.10; 0.22)	0.00002

Se: Sensibility, Sp: Specificity, PPV: Positive predictive value, NPV: Negative predictive value, PLR: Positive likelihood ratio, NLR: Negative likelihood ratio.

## Data Availability

The datasets generated and analyzed in the current study are not publicly available due to ethical restrictions on sharing a dataset because the data contain potentially identifying information. Further description or analysis of data are available from the authors upon reasonable request.

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
