# Peer review of "Accuracy of a Prehospital Triage Protocol in Predicting In-Hospital Mortality and Severe Trauma Cases among Older Adults"

_ijerph, 2023, doi:10.3390/ijerph20031975_

Round 1

Reviewer 1 Report

Very interesting study based on a large group of patients. The issues of aging populations and the provision of optimal emergency care are extremely important from a practical point of view. Identification of patients at risk of poor prognosis and risk of death is particularly important. The paper presented for review has a number of advantages. The authors present the emergency care system in use in France and also the criteria for the qualification of patients. It was based on a large group of patients, a retrospective analysis was made, and statistical analysis of individual elements associated with a worse prognosis in a group of elderly patients was performed.

I propose to consider several amendments or changes

1 I propose to expand the limitations including, in particular, the retrospective nature of the analysis and the need for further research

2 I suggest checking the p-value in Table 1 for each type of injury. Whether the p-value<0.001 applies to each type of injury or all together in my opinion needs to be detailed

3. the paper needs to be based on more literature

4. I suggest a broader analysis of other scales that are used in similar cases and to include in the discussion

Author Response

  1. This has been completed as suggested. L437-448
  2. Further analyses have been conducted to apply a p-value for each type.
  3. Additional sources have been added and developed to the manuscript in the introduction and discussion sections.
  4. The discussion has been improved accordingly. The results have been compared to a similar EMS (France) and others paramedic-led in other countries. L313-334

Reviewer 2 Report

I would like to thank the authors for this interesting work. The article is well-written, and the methodology is largely clear. It is also good to see that the authors are aware of possible limitations. I only have a few points to consider, please

(1)

It should be generally useful if the introduction may give further background on comparable protocols or other tools used in the triage context.

(2)

With the increasing adoption of Machine Learning (ML) / AI in the context under consideration, I find it necessary to refer to part of such contributions. The introduction can include examples of some applications in this regard, for example:

https://doi.org/10.3390/diagnostics12020241

https://doi.org/10.1109/BigData50022.2020.9378073

(3)

Given the large number of study population (8,888), I believe it would be interesting to consider using ML methods in future work. Perhaps ML-based predictions could be integrated with the protocol proposed.

Author Response

We would like to thank the reviewer 3 for his/her time and insightful comments. We have revised and improved the manuscript based on his/her feedback. Revisions are highlighted in the manuscript and are briefly summarized below. We hope that the modifications are sufficient to make our manuscript acceptable for publication

  1. The introduction has been completed. Comparison to other triage protocols/tools has been further developed in the discussion section as suggested by reviewer 1. L313-334
  2. Thank you for this very interesting suggestion. Accordingly, the potential impact and interest of ML/AI has been developed in the discussion section as future research pathways to improve the quality of studies based on large cohorts such as the one herein. Other references related to trauma patients have been cited. However, if you feel like the two ones you have mentioned are more appropriate, we would be pleased to add them. L450-464
  3. ML/AI has not been developed in the present study but will be considered for future work given the large number of patients in our registry.